# Relationship between dietary thiamine, riboflavin, and niacin intake and hypertension subtypes: A cross-sectional study from the 1999–2023

Siqi Ma[1], Qu Jin[2], Keying Yu[2], Hezeng Dong[1], Yanmei Lu[1], Liping Chang[2]*

**1** Changchun University of Traditional Chinese Medicine, Changchun City, Jilin Province, China,
**2** Changchun University of Traditional Chinese Medicine Affiliated Hospital, Changchun City, Jilin Province, China

* 15834807295@163.com

## Abstract

### Background

This cross-sectional study aimed to investigate whether vitamin B intake (thiamine, riboflavin, niacin) influences different hypertension subtypes and which subtype exhibits a more profound effect. The objective was to determine whether different vitamins should be supplemented according to specific hypertension subtypes.

### Methods

This cross-sectional study encompassed data from twenty-five years spanning 1999–2023, focusing on individuals with hypertension who possessed complete 24-hour dietary intake records and clinical assessments. Their total intake was disaggregated into two dimensions—'food sources' and 'supplement sources'—for separate estimation. Multivariable logistic regression models were employed to determine the odds ratios (OR) and 95% confidence intervals (95% CI) for associations between thiamine (vitamin B1), riboflavin (vitamin B2), and niacin (vitamin B3) intake and hypertension subtypes in subjects with hypertension versus those without. The relationship between nutrient intake and hypertension subtype risk was assessed using multivariable logistic regression models. To balance baseline characteristics, propensity score weighting was applied, followed by logistic regression. Logistic regression identified inflection points, with threshold effects assessed through piecewise logistic analysis. Inflection points were verified using likelihood ratio tests and bootstrap resampling. Heterogeneity and interactions between subgroups were evaluated via logistic regression models and likelihood ratio tests, respectively.

### Results

Analysis of NHANES data revealed significant demographic and dietary pattern differences between distinct hypertension subtypes. Multivariate regression analysis

**Data availability statement:** All relevant data are within the paper and its Supporting information files.

**Funding:** National Key Research and Development Programme of China (Project No.: 2019YFC1709900).

**Competing interests:** The authors have declared that no competing interests exist.

indicated that the initial positive association between dietary B vitamins and isolated systolic hypertension (ISH) largely disappeared upon adjustment, suggesting confounding effects from covariates such as age and comorbidities. Conversely, a strong independent positive association was observed between high riboflavin intake and systolic-diastolic hypertension (SDH). In the fully adjusted model, each unit increase in riboflavin was associated with a 25% increase in SDH risk (OR=1.25, 95% CI 1.05–1.49), exhibiting a significant dose-response trend. A potential threshold effect was observed near a 6 mg/day dose, beyond which the risk increment levelled off. Thiamine exhibited a potential non-linear association with SDH risk across quartiles, whereas no significant association was observed for niacin. Subgroup analyses indicated that the association between the three B vitamins and ISH risk was stronger among non-smokers and was influenced by gender (thiamine) and BMI (niacin).

## Conclusions

This study reveals subtype-specific hypertension risk factors. Whilst crude analyses indicated an association between B-vitamin intake and hypertension, multivariate adjustment revealed only riboflavin exhibited a significant independent positive correlation with the SDH subtype, suggesting it may represent a unique dietary-related factor for this subtype. The initial association with the ISH subtype was primarily attributable to confounding factors. The findings underscore the necessity for comprehensive adjustment and subtype stratification in nutritional epidemiology. The identified interaction with smoking status indicates that lifestyle factors play a crucial moderating role in dietary risk associations. The plateauing of SDH risk at high riboflavin levels suggests the need to further investigate potential threshold effects. These findings deepen our understanding of the relationship between diet and hypertension, emphasising that associations are not universally applicable across all manifestations of hypertension but are context-dependent. These findings suggest a complex relationship between riboflavin intake and hypertension subtypes, underscoring the need for personalized dietary approaches rather than universal supplementation. This study highlights the importance of maintaining balanced nutrition for cardiovascular health.

## Introduction

Hypertension constitutes a major health issue threatening public health systems worldwide. It is a significant risk factor for ischaemic heart disease, other cardiovascular diseases, stroke, and chronic kidney disease, affecting approximately 1.28 billion people globally [1,2]. In the United States, nearly half of all adults (48.1%, approximately 119.9 million people) suffer from hypertension [3]. Among 184,050 annual deaths, hypertension accounted for 30% of fatalities, resulting in 498,052 years of life lost [4]. Recent studies indicate that despite the regular and effective use

of antihypertensive medications, blood pressure control rates and therapeutic outcomes remain suboptimal, necessitating improvements in prevention, management, and treatment [5]. Therefore, identifying preventable risk factors and reducing cardiovascular risk and mortality through the prevention of hypertension are of paramount importance to global health.

Factors contributing to the development of hypertension include dietary habits, behavioural patterns, genetic predisposition, physiological factors, as well as socio-economic circumstances [6,7]. Diet plays a crucial role in preventing a range of chronic diseases, such as hypertension, cardiovascular disease, cancer, and type 2 diabetes [8,9]. Dietary intake ensures the smooth functioning of our physiological processes and maintains our overall health; its role is undeniably significant. Extensive research has revealed how specific dietary patterns and nutritional components can trigger or exacerbate the clinical symptoms of hypertension, and even prolong periods of disease remission [10]. Among the South Korean population, daily thiamine intake from food sources was associated with a reduced risk of hypertension (OR 0.95; 95% confidence interval 0.90–0.99) [11]. In hypertensive patients, targeted riboflavin supplementation in homozygous individuals (MTHFR 677TT genotype) reduces systolic blood pressure independently of antihypertensive medication [12]. Increasing evidence suggests that riboflavin exerts a novel role in regulating blood pressure (BP) among individuals with the MTHFR 677TT genotype based on these findings. Among Chinese adults, dietary niacin intake exhibits a J-shaped association with new-onset hypertension, with an inflection point at 15.6 mg/day. The lowest risk was observed for dietary niacin intakes ranging from 14.3 to 16.7 mg/day (third quartile) [13]. Among obese participants taking antihypertensive medication, higher riboflavin intake was associated with smaller increases in systolic blood pressure and pulse pressure [7].

However, existing research has not comprehensively examined and compared the relationship between the intake of these three key B vitamins (thiamine, riboflavin, and niacin) and the risk of isolated systolic hypertension (ISH), isolated diastolic hypertension (IDH), or combined systolic-diastolic hypertension (SDH) within a single, large, representative US adult population. US mortality data (CDC 2022) indicates that 518,000 patients die from hypertension annually. The underlying mechanisms for any subtype-specific associations remain unclear. The analysis of isolated systolic hypertension (ISH), isolated diastolic hypertension (IDH), and systolic-diastolic hypertension (SDH) reveals distinct vascular phenotypes: ISH reflects increased aortic stiffness and reduced renin levels, IDH indicates heightened sympathetic activity, while SDH suggests early endothelial injury. These subtypes exhibit distinct long-term risks for heart failure with preserved ejection fraction (HFpEF), chronic kidney disease (CKD), and stroke. By deciphering their molecular characteristics—such as COL1A1 methylation in ISH and ADRA1B gene variants in IDH—phenotype-specific application of renin-angiotensin system inhibitors, nebivolol, or sodium-glucose cotransporter 2 inhibitors can maximise the benefit-risk ratio. Existing research has not examined the relationship between vitamin B intake and isolated systolic hypertension, isolated diastolic hypertension, or combined systolic-diastolic hypertension in a US adult population. The underlying mechanisms remain unclear. This study further aims to systematically evaluate the association between vitamin B intake levels and the risk of developing different hypertension subtypes. Utilizing large-scale cohort data and rigorous statistical strategies, we endeavor to clarify the potential value of the vitamin B spectrum in the prevention and prognostic assessment of hypertension subtypes.

## Materials and methods

### Study population

This cross-sectional analysis utilised publicly available data from twenty-five years of the National Health and Nutrition Examination Survey (NHANES) spanning 1999–2023; all original materials were collected and collated by the Centers for Disease Control and Prevention (CDC) before being submitted to the National Center for Health Statistics (NCHS) for publication. Detailed information regarding NHANES survey methodology and analytical guidelines is accessible online at [NHANES Survey Methodology and Analytical Guidelines](https://wwwn.cdc.gov/nchs/nhanes/AnalyticGuidelines.aspx). The NHANES database is reviewed and managed by the NCHS Committee. Secondary analyses using this dataset do

not require additional ethical approval, as all participants provided voluntary medical consent. The primary website of the Department of Health and Human Services is accessible at [Centres for Disease Control and Prevention, Department of Health and Human Services](http://www.cdc.gov/nchs/nhanes.htm).

Participants under the age of 20 and those with incomplete data on thiamine intake, riboflavin intake, dietary niacin intake, hypertension subtype, or related covariates were excluded from the analysis. A total of 119,555 participants completed interviews across twenty-five years. Of these, 11,867 participants under 20 years of age, 25,195 participants with missing data on thiamine intake, riboflavin intake, or dietary niacin intake, 25,955 participants with missing hypertension subtype data, and 39,054 participants with incomplete covariate data were excluded. The final analysis sample comprised 17,484 participants. Among these, 3,591 subjects had hypertension, while 13,893 did not. Participants with isolated systolic hypertension numbered 2,521, those with isolated diastolic hypertension numbered 350, and those with combined systolic-diastolic hypertension numbered 720. A detailed flow diagram of the participant selection process is shown in Fig 1.

## Variables

**Nutrient intake.** In this study, the micronutrient category was restricted to water-soluble B vitamins, specifically comprising thiamine (vitamin B1), riboflavin (vitamin B2), and niacin (vitamin B3). Their total intake was disaggregated into two dimensions—'food sources' and 'supplement sources'—for separate estimation. The total daily intake for each nutrient (thiamine, riboflavin, niacin) was calculated and analyzed independently for its association with hypertension

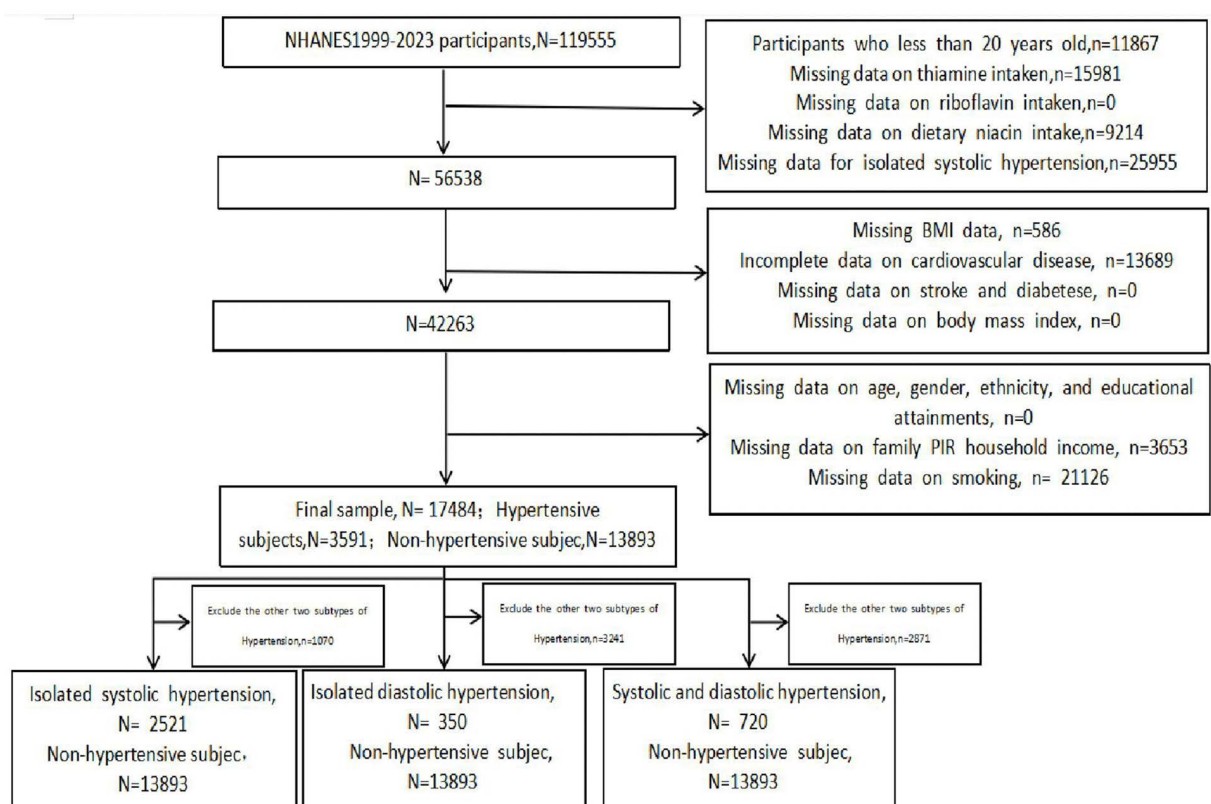

**Fig 1. Flow chart of study participants.** N, the number of patients being included; n, the number of patients being excluded.

subtypes. Participants contributed to the analysis for each vitamin based on their calculated total intake of that specific vitamin, regardless of the intake levels of the other B vitamins. Therefore, all participants were included in the analyses for all three vitamins. Twenty-four-hour dietary recall data were collected using the Computer-Assisted Dietary Interview (CADI) system from the National Health and Nutrition Examination Survey (NHANES). This tool employs a five-stage, multi-channel recall procedure supplemented by standardised prompting scripts to minimise recall bias [14]. Based on the Food and Nutrient Database for Dietary Studies (FNDDS) released concurrently with NHANES, we retrieved the energy and micronutrient density of all edible portions reported by respondents. Dietary intakes of thiamine, riboflavin, and niacin were then calculated according to actual consumption weights. During Mobile Examination Centre (MEC) interviews, participants recalled all vitamin/mineral supplement usage over the preceding 30 days. For each product, we extracted: frequency of intake (times/day or times/week), labelled active ingredient and unit dosage, and number of tablets/ capsules/drops consumed per intake. Multiplying these parameters and dividing by 30 yielded individual daily supplement contribution values. The dietary and supplemental contributions for each nutrient were summed to yield the individual's total daily intake (mg d$^{-1}$). Participants were subsequently assigned to four increasing quartile groups (Q1–Q4) based on the distribution quartiles for thiamine, riboflavin, and niacin, facilitating subsequent dose–response analysis.

**Hypertension subtypes.** Hypertension was defined as a mean systolic blood pressure (SBP) ≥140 mmHg, or a mean diastolic blood pressure (DBP) ≥90 mmHg, or self-reported current use of antihypertensive medication. Based on the pathogenesis of hypertension, it is categorised into three subtypes: 1. Isolated systolic hypertension (ISH), defined as mean systolic blood pressure (SBP) ≥140 mmHg and mean diastolic blood pressure (DBP) < 90 mmHg; 2. Isolated diastolic hypertension (IDH), defined as mean systolic blood pressure (SBP) < 140 mmHg and mean diastolic blood pressure (DBP) ≥90 mmHg; 3. Systolic-diastolic hypertension, defined as mean systolic blood pressure (SBP) ≥140 mmHg and mean diastolic blood pressure (DBP) ≥90 mmHg [15–17]. For participants on antihypertensive medication, the subtype classification was based on their measured BP values. The method for calculating mean systolic and diastolic blood pressure involves taking three measurements of systolic and diastolic pressure at five-minute intervals from each subject, as recorded in the NHANSE database, and calculating the mean value.

**Covariables.** To mitigate confounding related to hypertension subtypes and pulse-pressure variability, we extracted a comprehensive covariate set from NHANES. Demographics included age, sex, race/ethnicity (Hispanic/Latino, non-Hispanic White, non-Hispanic Black, other), education (< 9, 9–12, ≥12 years) and household income-to-poverty ratio (low ≤ 1.3, moderate 1.3–3.5, high > 3.5). Clinical variables comprised physician-diagnosed coronary heart disease, diabetes mellitus or stroke, body-mass index (kg m$^{-2}$) and smoking status (never < 100 lifetime cigarettes, former ≥ 100 quit, current). Total energy intake (kcal), Potassium and sodium intake (mg/d) was also included as a covariate.

**Statistical analysis.** Survey Design and Weighting: All analyses accounted for the complex, multistage, probability sampling design of NHANES by incorporating survey weights, strata, and primary sampling units (PSUs). The two-day dietary sample weights (WTDR2D) for each cycle were used and combined according to NHANES analytical guidelines to create a normalized weight for the 12-cycle dataset. All descriptive statistics and regression models were performed using the 'survey' package in R (version 4.2.1) to produce nationally representative estimates. Categorical data are reported as unweighted counts (weighted proportions); continuous data as weighted means ± standard errors. The assessment of group differences was conducted through the utilisation of linear regression (continuous) and chi-squared tests (categorical). The estimation of odds ratios (OR) and 95% confidence intervals (CI) for hypertension subtypes across quartiles of thiamine (B1), riboflavin (B2) and niacin (B3) was conducted using three nested logistic models. The first model (Model 1) incorporates age, sex, race/ethnicity, education, the poverty-income ratio. The second model (Model 2) adds diabetes, coronary heart disease, and stroke. The third model (Model 3) includes BMI, smoking, total energy intake, Potassium intake and sodium intake. Trend tests employed quartile medians; sensitivity analyses were conducted on all models. A two-piece logistic model was employed to identify intake thresholds, and bootstrap resampling was utilised to verify breakpoints. The consistency of the data was assessed through interaction and subgroup analyses (sex, ethnicity,

BMI ≥ 25 kg m$^{-2}$). The heterogeneity of the data was evaluated using logistic regression and likelihood-ratio tests. The analyses were conducted in R 4.2.1 (survey package 4.1-1). *A fully anonymized version of the dataset used in this study is provided as* S2 Table.

## Results

### The study population comprised the following subjects

This real-time study ultimately included 17,484 participants from the National Health and Nutrition Examination Survey (NHANES) between 1999 and 2023. The detailed inclusion and exclusion process is illustrated in Fig 1.

### Baseline characteristics

Table 1 presents baseline characteristics of the study population by hypertension subtype, comparing each with the non-hypertensive cohort. Analyses encompassed three groups: isolated systolic hypertension (ISH, n = 2,521), isolated diastolic hypertension (IDH, n = 350), and mixed systolic-diastolic hypertension (SDH, n = 720). Significant differences existed between groups. Compared with the non-hypertensive cohort, participants in the ISH group were significantly older (mean age 63.6 years vs 48.9 years, p < 0.001), had lower mean dietary intakes of thiamine, riboflavin, niacin, potassium, sodium, and total energy (all p < 0.05), and exhibited higher prevalence rates of diabetes mellitus, coronary heart disease, and stroke. In contrast, the IDH group comprised younger individuals (mean age 47.4 years vs 50.6 years, p = 0.002), a significantly higher proportion of males (67.8% vs 54.8%, p < 0.001), higher mean sodium intake (p = 0.009), and markedly higher obesity prevalence (BMI > 30 kg/m$^2$: 57.1% vs. 36.0%, p < 0.001). The SDH group exhibited a centralised age distribution (mean 53.6 years, p < 0.001), significantly lower riboflavin intake, higher sodium intake (p < 0.001), and a higher proportion of non-Hispanic Black individuals (19.3% vs. 8.9%, p < 0.001). Significant differences in educational attainment, household PIR, and smoking status were also observed across hypertensive subtypes.

### Single-factor analysis

Table 2 presents the results of univariate logistic regression analyses examining factors associated with different hypertension subtypes. B-vitamin intake exhibited significant yet modest positive correlations with specific subtypes: thiamine (odds ratio 1.17, 95% confidence interval 1.11–1.23) and riboflavin (odds ratio 1.16, 95% confidence interval 1.12–1.21) were both associated with increased risk of isolated systolic hypertension (ISH). Riboflavin also showed a positive association with systolic-diastolic hypertension (SDH) (OR=1.21, 95% CI 1.13–1.30). Several demographic and clinical factors exhibited distinct association patterns across subtypes: increasing age provided significant protection against ISH and SDH but constituted a risk factor for isolated diastolic hypertension (IDH). Female gender was associated with increased risk of IDH (OR=1.58) and SDH (OR=1.32). Higher educational attainment (>12 years) was associated with increased risk in both ISH (OR 2.30) and SDH (OR 1.52), but with reduced risk in IDH (OR 0.49). Non-Hispanic Black ethnicity was associated with significantly reduced risk across all subtypes, most markedly for SDH (OR 0.30). Absence of coronary heart disease and stroke was strongly associated with significantly increased risk of ISH. Obesity (BMI > 30 kg/m$^2$) was a strong protective factor for both IDH (OR = 0.34) and SDH (OR = 0.65). Dietary sodium intake showed no significant association with SDH, and potassium intake also showed no significant association with either IDH or SDH in this unadjusted analysis. These findings highlight phenotype-specific determinants, supporting a stratified approach in hypertension prevention and management(as presented in Table 2).

### Multivariate analysis

As shown in Table 3, the initially observed positive association between dietary B vitamin intake and isolated systolic hypertension (ISH) largely diminished after comprehensive multivariable adjustment. When analysed as continuous

**Table 1. Population characteristics categorised by isolated systolic hypertension, isolated diastolic hypertension, and combined systolic-diastolic hypertension.**

| Variables | Total | Subjects with isolated systolic hypertension | Non-hypertensive subjec | P value | Total | Subjects with Isolated diastolic hypertension | Non-hypertensive subjec | p value | Total | Subjects with Systolic and diastolic hypertension | Non-hypertensive subjec | p value |
|---|---|---|---|---|---|---|---|---|---|---|---|---|
| NO. | 16414 | 2521 | 13893 | | 14243 | 350 | 13893 | | 14613 | 720 | 13893 | |
| Thiamine, mean(SE), mg/d | 1.61 (0.01) | 1.51 (0.02) | 1.62 (0.01) | <0.001 | 1.61 (0.01) | 1.67 (0.07) | 1.61 (0.01) | 0.47 | 1.61 (0.01) | 1.59 (0.06) | 1.61 (0.01) | 0.003 |
| Riboflavin, mean(SE), mg/d | 2.23 (0.02) | 2.05 (0.03) | 2.25 (0.02) | <0.001 | 2.23 (0.02) | 2.28 (0.10) | 2.23 (0.02) | 0.98 | 2.23 (0.02) | 2.04 (0.07) | 2.24 (0.02) | < 0.001 |
| Niacin, mean (SE), mg/d | 25.49 (0.20) | 23.57 (0.45) | 25.72 (0.20) | <0.001 | 25.49 (0.20) | 26.19 (1.13) | 25.47 (0.20) | 0.83 | 25.49 (0.20) | 24.84 (0.92) | 25.51 (0.20) | 0.24 |
| Age, mean (SE) | 50.53 (0.23) | 63.61 (0.41) | 48.93 (0.23) | <0.001 | 50.53 (0.23) | 47.40 (1.00) | 50.60 (0.24) | 0.002 | 50.53 (0.23) | 53.62 (0.65) | 50.41 (0.24) | < 0.001 |
| Gender, n (%) | | | | 0.044 | | | | < 0.001 | | | | 0.14 |
| Male | 9,983 (55.04%) | 1,457 (51.95%) | 8,526 (55.42%) | | 9,983 (55.04%) | 235 (67.81%) | 9,748 (54.75%) | | 9,983 (55.04%) | 454 (59.57%) | 9,529 (54.88%) | |
| Female | 7,501 (44.96%) | 1,064 (48.05%) | 6,437 (44.58%) | | 7,501 (44.96%) | 115 (32.19%) | 7,386 (45.25%) | | 7,501 (44.96%) | 266 (40.43%) | 7,235 (45.12%) | |
| Race, n (%) | | | | <0.001 | | | | 0.038 | | | | <0.001 |
| Non-Hispanic white | 9,624 (74.57%) | 1,354 (75.89%) | 8,270 (74.40%) | | 9,624 (74.57%) | 179 (69.21%) | 9,445 (74.69%) | | 9,624 (74.57%) | 278 (65.38%) | 9,346 (74.90%) | |
| Non-Hispanic black | 3,275 (9.24%) | 603 (12.53%) | 2,672 (8.84%) | | 3,275 (9.24%) | 81 (11.92%) | 3,194 (9.18%) | | 3,275 (9.24%) | 277 (19.34%) | 2,998 (8.88%) | |
| Mexican American | 2,230 (6.01%) | 306 (4.26%) | 1,924 (6.22%) | | 2,230 (6.01%) | 37 (4.85%) | 2,193 (6.04%) | | 2,230 (6.01%) | 72 (5.78%) | 2,158 (6.02%) | |
| Others | 2,355 (10.18%) | 258 (7.33%) | 2,097 (10.53%) | | 2,355 (10.18%) | 53 (14.02%) | 2,302 (10.09%) | | 2,355 (10.18%) | 93 (9.49%) | 2,262 (10.20%) | |
| Education Level, n (%) | | | | 0.001 | | | | 0.180 | | | | 0.045 |
| < 9 | 1,611 (4.85%) | 394 (9.46%) | 1,217 (4.29%) | | 1,611 (4.85%) | 15 (2.50%) | 1,596 (4.90%) | | 1,611 (4.85%) | 75 (5.37%) | 1,536 (4.83%) | |
| 9–12 | 7,525 (41.42%) | 1,089 (42.91%) | 6,436 (41.23%) | | 7,525 (41.42%) | 150 (40.12%) | 7,375 (41.45%) | | 7,525 (41.42%) | 346 (47.38%) | 7,179 (41.20%) | |
| >12 | 8,341 (53.74%) | 1,037 (47.63%) | 7,304 (54.48%) | | 8,341 (53.74%) | 185 (57.38%) | 8,156 (53.65%) | | 8,341 (53.74%) | 299 (47.25%) | 8,042 (53.97%) | |
| Family PIR infmpir, n (%) | | | | 0.001 | | | | 0.049 | | | | < 0.001 |
| Low | 3,659 (15.86%) | 484 (14.76%) | 3,175 (15.99%) | | 3,659 (15.86%) | 78 (19.92%) | 3,581 (15.76%) | | 3,659 (15.86%) | 195 (19.61%) | 3,464 (15.72%) | |
| Medium | 7,617 (38.20%) | 1,255 (43.86%) | 6,362 (37.51%) | | 7,617 (38.20%) | 134 (29.66%) | 7,483 (38.39%) | | 7,617 (38.20%) | 319 (39.58%) | 7,298 (38.15%) | |
| High | 6,208 (45.95%) | 782 (41.38%) | 5,426 (46.50%) | | 6,208 (45.95%) | 138 (50.42%) | 6,070 (45.85%) | | 6,208 (45.95%) | 206 (40.81%) | 6,002 (46.13%) | |

*(Continued)*

Table 1. (Continued)

| Variables | Total | Subjects with isolated systolic hypertension | Non-hypertensive subjec | P value | Total | Subjects with Isolated diastolic hypertension | Non-hypertensive subjec | p value | Total | Subjects with Systolic and diastolic hypertension | Non-hypertensive subjec | p value |
|---|---|---|---|---|---|---|---|---|---|---|---|---|
| Diabetes, n (%) | | | | <0.001 | | | | 0.860 | | | | 0.019 |
| Yes | 2,450 (10.49%) | 560 (17.98%) | 1,890 (9.58%) | | 2,450 (10.49%) | 38 (10.98%) | 2,412 (10.48%) | | 2,450 (10.49%) | 112 (12.43%) | 2,338 (10.42%) | |
| No | 15,025 (89.51%) | 1,959 (82.02%) | 13,066 (90.42%) | | 15,025 (89.51%) | 312 (89.02%) | 14,713 (89.52%) | | 15,025 (89.51%) | 608 (87.57%) | 14,417 (89.58%) | |
| Coronary heart disease, n (%) | | | | <0.001 | | | | 0.010 | | | | 0.450 |
| Yes | 1,056 (4.96%) | 263 (8.41%) | 793 (4.54%) | | 1,056 (4.96%) | 8 (1.39%) | 1,048 (5.04%) | | 1,056 (4.96%) | 35 (3.78%) | 1,021 (5.00%) | |
| No | 16,350 (94.75%) | 2,241 (91.15%) | 14,109 (95.19%) | | 16,350 (94.75%) | 342 (98.61%) | 16,008 (94.66%) | | 16,350 (94.75%) | 682 (95.76%) | 15,668 (94.71%) | |
| Stroke, n (%) | | | | <0.001 | | | | 0.052 | | | | 0.440 |
| Yes | 865 (3.56%) | 226 (6.77%) | 639 (3.17%) | | 865 (3.56%) | 23 (6.26%) | 842 (3.50%) | | 865 (3.56%) | 46 (4.40%) | 819 (3.53%) | |
| No | 16,590 (96.30%) | 2,285 (92.86%) | 14,305 (96.72%) | | 16,590 (96.30%) | 327 (93.74%) | 16,263 (96.36%) | | 16,590 (96.30%) | 672 (95.37%) | 15,918 (96.33%) | |
| SMOKE, n (%) | | | | <0.001 | | | | 0.550 | | | | 0.280 |
| Current | 7,539 (44.36%) | 767 (30.93%) | 6,772 (46.00%) | | 7,539 (44.36%) | 161 (42.11%) | 7,378 (44.41%) | | 7,539 (44.36%) | 350 (47.21%) | 7,189 (44.26%) | |
| Former | 9,944 (55.64%) | 1,754 (69.07%) | 8,190 (54.00%) | | 9,944 (55.64%) | 189 (57.89%) | 9,755 (55.59%) | | 9,944 (55.64%) | 370 (52.79%) | 9,574 (55.74%) | |
| Body mass index, n (%), kg/m² | | | | 0.430 | | | | < 0.001 | | | | < 0.001 |
| < 25 | 4,925 (29.41%) | 691 (27.89%) | 4,234 (29.60%) | | 4,925 (29.41%) | 54 (14.12%) | 4,871 (29.76%) | | 4,925 (29.41%) | 166 (20.76%) | 4,759 (29.73%) | |
| 25-30 | 5,957 (34.17%) | 889 (35.61%) | 5,068 (33.99%) | | 5,957 (34.17%) | 95 (28.77%) | 5,862 (34.29%) | | 5,957 (34.17%) | 226 (32.77%) | 5,731 (34.22%) | |
| >30 | 6,602 (36.42%) | 941 (36.50%) | 5,661 (36.41%) | | 6,602 (36.42%) | 201 (57.11%) | 6,401 (35.95%) | | 6,602 (36.42%) | 328 (46.47%) | 6,274 (36.06%) | |
| potassium, mean (SE), mg/d | 2,722.71 (16.54) | 2,616.04 (37.65) | 2,735.72 (17.93) | 0.008 | 2,722.71 (16.54) | 2,926.65 (110.03) | 2,718.07 (16.68) | 0.140 | 2,722.71 (16.54) | 2,659.71 (76.14) | 2,724.99 (17.14) | 0.390 |
| sodium, mean (SE), mg/d | 3,526.52 (21.14) | 3,214.71 (50.39) | 3,564.53 (23.10) | <0.001 | 3,526.52 (21.14) | 4,017.83 (160.63) | 3,515.32 (21.30) | 0.009 | 3,526.52 (21.14) | 3,566.72 (100.99) | 3,525.07 (21.69) | 0.740 |
| Energy, mean (SE), kcal | 2,208.56 (10.65) | 1,991.38 (26.65) | 2,235.03 (11.86) | <0.001 | 2,208.56 (10.65) | 2,485.92 (87.24) | 2,202.24 (10.68) | < 0.001 | 2,208.56 (10.65) | 2,247.45 (59.82) | 2,207.16 (11.08) | 0.760 |

**Table 2. Correlation between covariates and the risk of isolated systolic hypertension, isolated diastolic hypertension, and combined systolic-diastolic hypertension.**

| Variable | Isolated systolic hypertension | | Isolated diastolic hypertension | | Systolic and diastolic hypertension | |
|---|---|---|---|---|---|---|
| | OR_95CI | P_value | OR_95CI | P_value | OR_95CI | P_value |
| Thiamine | 1.17 (1.11~1.23) | <0.001 | 0.97 (0.87~1.08) | 0.526 | 1.14 (1.04~1.25) | 0.003 |
| Riboflavin | 1.16 (1.12~1.21) | <0.001 | 0.96 (0.9~1.03) | 0.279 | 1.21 (1.13~1.3) | <0.001 |
| Niacin | 1.01 (1.01~1.02) | <0.001 | 1 (0.99~1) | 0.206 | 1 (1~1.01) | 0.073 |
| Age | 0.94 (0.94~0.94) | <0.001 | 1.01 (1.01~1.02) | <0.001 | 0.98 (0.98~0.99) | <0.001 |
| Gender, n (%) | | | | | | |
| Male | 1(Ref) | | 1(Ref) | | 1(Ref) | |
| Female | 1.06 (0.97~1.15) | 0.197 | 1.58 (1.26~1.98) | <0.001 | 1.32 (1.13~1.54) | <0.001 |
| Education. Level, n (%) | | | | | | |
| < 9 | 1(Ref) | | 1(Ref) | | 1(Ref) | |
| 9–12 | 1.91 (1.67~2.18) | <0.001 | 0.53 (0.31~0.9) | 0.019 | 1.14 (0.88~1.48) | 0.311 |
| >12 | 2.3 (2.01~2.62) | <0.001 | 0.49 (0.29~0.83) | 0.008 | 1.52 (1.17~1.97) | 0.002 |
| Family PIR infmpir, n (%) | | | | | | |
| Low | 1(Ref) | | 1(Ref) | | 1(Ref) | |
| Medium | 0.79 (0.7~0.88) | <0.001 | 1.19 (0.89~1.57) | 0.239 | 1.24 (1.04~1.5) | 0.019 |
| High | 1.08 (0.96~1.22) | 0.196 | 0.99 (0.75~1.31) | 0.943 | 1.66 (1.36~2.03) | <0.001 |
| Race, n (%) | | | | | | |
| Non-Hispanic white | 1(Ref) | | 1(Ref) | | 1(Ref) | |
| Non-Hispanic black | 0.67 (0.6~0.74) | <0.001 | 0.65 (0.5~0.85) | 0.002 | 0.3 (0.25~0.35) | <0.001 |
| Mexican American | 1.03 (0.9~1.18) | 0.688 | 1.12 (0.79~1.61) | 0.522 | 0.9 (0.69~1.17) | 0.42 |
| Others | 1.31 (1.14~1.51) | <0.001 | 0.84 (0.62~1.15) | 0.282 | 0.75 (0.59~0.95) | 0.017 |
| Coronary heart disease, n (%) | | | | | | |
| Yes | 1(Ref) | | 1(Ref) | | 1(Ref) | |
| No | 2.05 (1.77~2.37) | <0.001 | 0.41 (0.2~0.83) | 0.013 | 0.9 (0.63~1.27) | 0.533 |
| Stroke, n (%) | | | | | | |
| Yes | 1(Ref) | | 1(Ref) | | 1(Ref) | |
| No | 2.31 (1.97~2.71) | <0.001 | 1.64 (1.07~2.53) | 0.024 | 1.6 (1.17~2.18) | 0.003 |
| Diabetes, n (%) | | | | | | |
| Yes | 1(Ref) | | 1(Ref) | | 1(Ref) | |
| No | 2 (1.79~2.22) | <0.001 | 0.85 (0.6~1.2) | 0.35 | 1.29 (1.04~1.58) | 0.018 |
| SMOKE, n (%) | | | | | | |
| Current | 1(Ref) | | 1(Ref) | | 1(Ref) | |
| Former | 0.53 (0.49~0.58) | <0.001 | 1.04 (0.84~1.28) | 0.73 | 1.15 (0.99~1.34) | 0.063 |
| Body mass index (kg/m2), n (%) | | | | | | |
| < 25 | 1(Ref) | | 1(Ref) | | 1(Ref) | |
| 25-30 | 0.92 (0.83~1.02) | 0.126 | 0.67 (0.48~0.94) | 0.021 | 0.87 (0.71~1.07) | 0.178 |
| >30 | 0.94 (0.84~1.04) | 0.246 | 0.34 (0.25~0.47) | <0.001 | 0.65 (0.53~0.78) | <0.001 |
| Energy, kcal | 1(1~1.02) | <0.001 | 1(1~1.04) | <0.001 | 1(1~1.06) | 0.504 |
| potassium, mg/d | 0.89(0.81~1.04) | 0.007 | 0.86(0.65~1.22) | 0.038 | 0.85 (0.62~1.43) | 0.426 |
| sodium, mg/d | 1.08 (0.84~1.20) | <0.001 | 1.08 (0.92~1.37) | <0.001 | 1.06 (0.91~1.42) | 0.683 |

**Table 3. Relationship between dietary thiamine, riboflavin, and niacin intake and isolated systolic hypertension among adult participants in the NHANES.**

| Quartiles | OR (95% CI) | | | | | | | | |
|---|---|---|---|---|---|---|---|---|---|
| | No. | Crude | p-Value | Model 1 | p-Value | Model 2 | p-Value | Model 3 | p-Value |
| **Thiamine** | 16414 | | | | | | | | |
| Thiamine cut1 (≤0.965) | 4056 | 1(Ref) | | 1(Ref) | | 1(Ref) | | 1(Ref) | |
| Thiamine cut2 (0.966–1.399) | 4123 | 1.12(0.96~1.30) | 0.156 | 1.09 (0.92~1.29) | 0.301 | 1.10 (0.93~1.29) | 0.285 | 1.09 (0.91~1.31) | 0.343 |
| Thiamine cut3 (1.400–1.974) | 4113 | 1.10 (0.95~1.28) | 0.212 | 1.02 (0.87~1.20) | 0.794 | 1.02 (0.87~1.20) | 0.78 | 1.02 (0.84~1.23) | 0.846 |
| Thiamine cut4 (≥13.083) | 4122 | 1.46 (1.20~1.78) | <0.001 | 1.11 (0.89~1.38) | 0.345 | 1.11 (0.810~1.48) | 0.341 | 1.10 (0.81~1.48) | 0.537 |
| Trend.test | | | <0.001 | | 0.472 | | 0.473 | | 0.676 |
| Thiamine | 16414 | 1.15(1.07~1.24) | <0.001 | 1.03(0.95~1.11) | 0.500 | 1.03(0.95~1.11) | 0.505 | 1.01(0.90~1.14) | 0.817 |
| **Riboflavin** | 16414 | | | | | | | | |
| Riboflavin cut1 (≤1.294) | 4045 | 1(Ref) | | 1(Ref) | | 1(Ref) | | 1(Ref) | |
| Riboflavin cut2 (1.295–1.887) | 4093 | 0.95 (0.80~1.12) | 0.544 | 0.96 (0.81~1.15) | 0.631 | 0.97 (0.82~1.16) | 0.618 | 0.98 (0.82~1.17) | 0.597 |
| Riboflavin cut3 (1.888–2.630) | 4144 | 1.13 (0.93~1.38) | 0.017 | 1.07 (0.87~1.33) | 0.019 | 1.08 (0.87~1.33) | 0.020 | 1.09 (0.88~1.36) | 0.021 |
| Riboflavin cut4 (≥2.631) | 4132 | 1.42 (1.18~1.70) | <0.001 | 1.12 (0.92~1.36) | 0.021 | 1.12 (0.92~1.37) | 0.025 | 1.15 (0.90~1.48) | 0.025 |
| Trend.test | | | <0.001 | | 0.024 | | 0.025 | | 0.025 |
| Riboflavin | 16414 | 1.15 (1.09~1.21) | <0.001 | 1.06 (1.00~1.11) | 0.017 | 1.06 (1.00~1.11) | 0.019 | 1.08 (1.11~1.16) | 0.022 |
| **Niacin** | 16414 | | | | | | | | |
| Niacin.cut1 (≤14.584) | 4095 | 1(Ref) | | 1(Ref) | | 1(Ref) | | 1(Ref) | |
| Niacin.cut2(14.585–21.376) | 4096 | 1.2 (0.99~1.46) | 0.062 | 1.21 (0.98~1.49) | 0.077 | 1.21 (0.98~1.49) | 0.073 | 1.21 (0.98~1.50) | 0.077 |
| Niacin.cut3 (21.377–222.552) | 4113 | 1.18 (1.00~1.39) | 0.049 | 1.03 (0.85~1.25) | 0.744 | 1.03 (0.85~1.25) | 0.74 | 1.03 (0.85~1.26) | 0.744 |
| Niacin.cut4(≥222.553) | 4110 | 1.55 (1.27~1.88) | <0.001 | 1.06 (0.85~1.32) | 0.599 | 1.06 (0.85~1.32) | 0.604 | 1.05 (0.83~1.34) | 0.683 |
| Trend.test | | | <0.001 | | 0.933 | | 0.94 | | 0.998 |
| Niacin | 16414 | 1.01 (1.01~1.02) | <0.001 | 1 (0.99~1.00) | 0.74 | 1 (0.99~1.00) | 0.755 | 1 (0.99~1.00) | 0.669 |

variables in the fully adjusted model (Model 3), thiamine (odds ratio 1.01, 95% confidence interval 0.90–1.14), riboflavin (OR 1.08, 95% CI 1.01–1.16), or niacin (OR 1.00, 95% CI 0.99–1.00) showed no statistically significant association with ISH risk. Quartile analyses yielded consistent results. For riboflavin, a marginally significant positive trend was observed across quartiles in Models 2 and 3 (trend p-value = 0.025), though no significant odds ratio was demonstrated between the highest and lowest quartiles. After full adjustment, no significant trends were observed for thiamine or niacin. These findings suggest the original association may be confounded by other demographic and clinical factors. Table 4 presents the results of the multivariate logistic regression analysis, which assessed the association between dietary intake of B vitamins and combined systolic-diastolic hypertension (SDH). Riboflavin demonstrated a significant positive association

**Table 4. Relationship between dietary thiamine, riboflavin, and niacin intake and systolic and diastolic hypertension among adult participants in the NHANES.**

| Quartiles | OR (95% CI) | | | | | | | | |
|---|---|---|---|---|---|---|---|---|---|
| | No. | Crude | p-Value | Model 1 | p-Value | Model 2 | p-Value | Model 3 | p-Value |
| **Thiamine** | 14613 | | | | | | | | |
| Thiamine cut1 (≤0.965) | 3572 | 1(Ref) | | 1(Ref) | | 1(Ref) | | 1(Ref) | |
| Thiamine cut2 (0.966–1.399) | 3622 | 1.38 (0.99~1.93) | 0.06 | 1.35 (0.95~1.91) | 0.089 | 1.35 (0.95~1.92) | 0.089 | 1.43 (1.00~2.04) | 0.052 |
| Thiamine cut3 (1.400–1.974) | 3651 | 1.43 (1.04~1.96) | 0.029 | 1.41 (1.01~1.96) | 0.045 | 1.40 (1.01~1.96) | 0.047 | 1.57 (1.09~2.25) | 0.015 |
| Thiamine cut4 (≥13.083) | 3768 | 1.36 (0.99~1.87) | 0.058 | 1.33 (0.95~1.86) | 0.098 | 1.33 (0.95~1.86) | 0.098 | 1.64 (1.11~2.41) | 0.012 |
| Trend.test | | | 0.066 | | 0.104 | | 0.104 | | 0.013 |
| Thiamine | 14613 | 1.02 (0.88~1.18) | 0.766 | 1.01 (0.87~1.18) | 0.887 | 1.01 (0.87~1.18) | 0.873 | 1.05 (0.85~1.30) | 0.634 |
| **Riboflavin** | 14613 | | | | | | | | |
| Riboflavin cut1 (≤1.294) | 3560 | 1(Ref) | | 1(Ref) | | 1(Ref) | | 1(Ref) | |
| Riboflavin cut2 (1.295–1.887) | 3574 | 1.14 (0.82~1.58) | 0.001 | 1.09 (0.78~1.53) | 0.023 | 1.27 (1.04~1.56) | 0.024 | 1.19 (0.85~1.68) | 0.026 |
| Riboflavin cut3 (1.888–2.630) | 3698 | 1.25 (0.90~1.73) | <0.001 | 1.17 (0.83~1.65) | 0.001 | 1.46 (1.18~1.82) | 0.001 | 1.35 (0.94~1.93) | 0.002 |
| Riboflavin cut4 (≥2.631) | 3781 | 1.57 (1.12~2.22) | <0.001 | 1.45 (1.00~2,10) | 0.005 | 1.4 (1.12~1.74) | 0.004 | 1.9 (1.27~2.85) | 0.005 |
| Trend.test | | | <0.001 | | 0.001 | | 0.001 | | 0.001 |
| Riboflavin | 14613 | 1.15 (1.02~1.29) | <0.001 | 1.12 (0.99~1.26) | 0.001 | 1.12 (0.99~1.26) | 0.001 | 1.25 (1.05~1.49) | 0.001 |
| **Niacin** | 14613 | | | | | | | | |
| Niacin.cut1 (≤14.584) | 3519 | 1(Ref) | | 1(Ref) | | 1(Ref) | | 1(Ref) | |
| Niacin.cut2(14.585–21.376) | 3646 | 0.91 (0.66~1.25) | 0.565 | 0.91 (0.66~1.25) | 0.576 | 0.91 (0.66~1.25) | 0.548 | 0.9 (0.65~1.24) | 0.514 |
| Niacin.cut3 (21.377–222.552) | 3652 | 1.16 (0.81~1.66) | 0.412 | 1.16 (0.82~1.65) | 0.495 | 1.16 (0.82~1.65) | 0.399 | 1.16 (0.81~1.65) | 0.42 |
| Niacin.cut4(≥222.553) | 3796 | 1.12 (0.80~1.56) | 0.512 | 1.09 (0.79~1.52) | 0.541 | 1.10 (0.79~1.52) | 0.58 | 1.11 (0.78~1.57) | 0.57 |
| Trend.test | | | 0.278 | | 0.292 | | 0.318 | | 0.316 |
| Niacin | 14613 | 1 (0.99~1.01) | 0.49 | 1 (0.99~1.01) | 0.619 | 1 (0.99~1.01) | 0.595 | 1 (1~1.01) | 0.554 |

across all adjusted models. In the fully adjusted model (Model 3), when analysed as a continuous variable, each unit increase in riboflavin intake was associated with a 25% higher risk of SDH (odds ratio 1.25, 95% confidence interval 1.05–1.49, p=0.001). Quartile analysis further confirmed this association, revealing a significant positive dose-response trend (trend p-value=0.001). Compared with the lowest intake quartile, participants in the highest intake quartile nearly doubled their odds of developing SDH (OR = 1.90, 95% CI 1.27–2.85, p=0.005). Conversely, the association for thiamine was evident only in the quartile analysis, with no significant association observed in the continuous variable analysis. In Model 3, subjects in the third quartile (OR = 1.57, 95% CI 1.09–2.25) and fourth quartile (OR = 1.64, 95% CI 1.11–2.41) exhibited significantly higher SDH risk than the first quartile, with a statistically significant trend test (p-trend=0.013). However, no significant association was observed for continuous thiamine variables in any model. Neither continuous variables nor quartile analyses revealed a significant association between dietary niacin intake and SDH risk across all adjusted models. These findings indicate a clear and strong positive correlation between riboflavin and SDH risk. It is noteworthy that after extensive adjustment for demographic, socioeconomic, and clinical confounders (Model 3), the positive correlation between riboflavin and SDH persisted, suggesting an independent relationship.

## Associations of vitamin intake from food and supplement sources with hypertension subtypes

To further elucidate the contributions of different sources, we separately analyzed the associations of vitamin intake derived solely from food and solely from supplements with the hypertension subtypes (S1 Table). For riboflavin, both food-derived and supplement-derived intake were positively associated with SDH risk in a dose-response manner. For ISH, the association was primarily driven by food-derived riboflavin. Thiamine and niacin from either source showed no

consistent or significant associations with any hypertension subtype, reinforcing the primary findings from the total intake analysis.

## Non-linear relationship

Fig 2 employs a restricted cubic spline model to illustrate the non-linear relationship between riboflavin intake and the risk of systolic-diastolic hypertension (SDH). The overall association is statistically significant (P = 0.002), whereas the

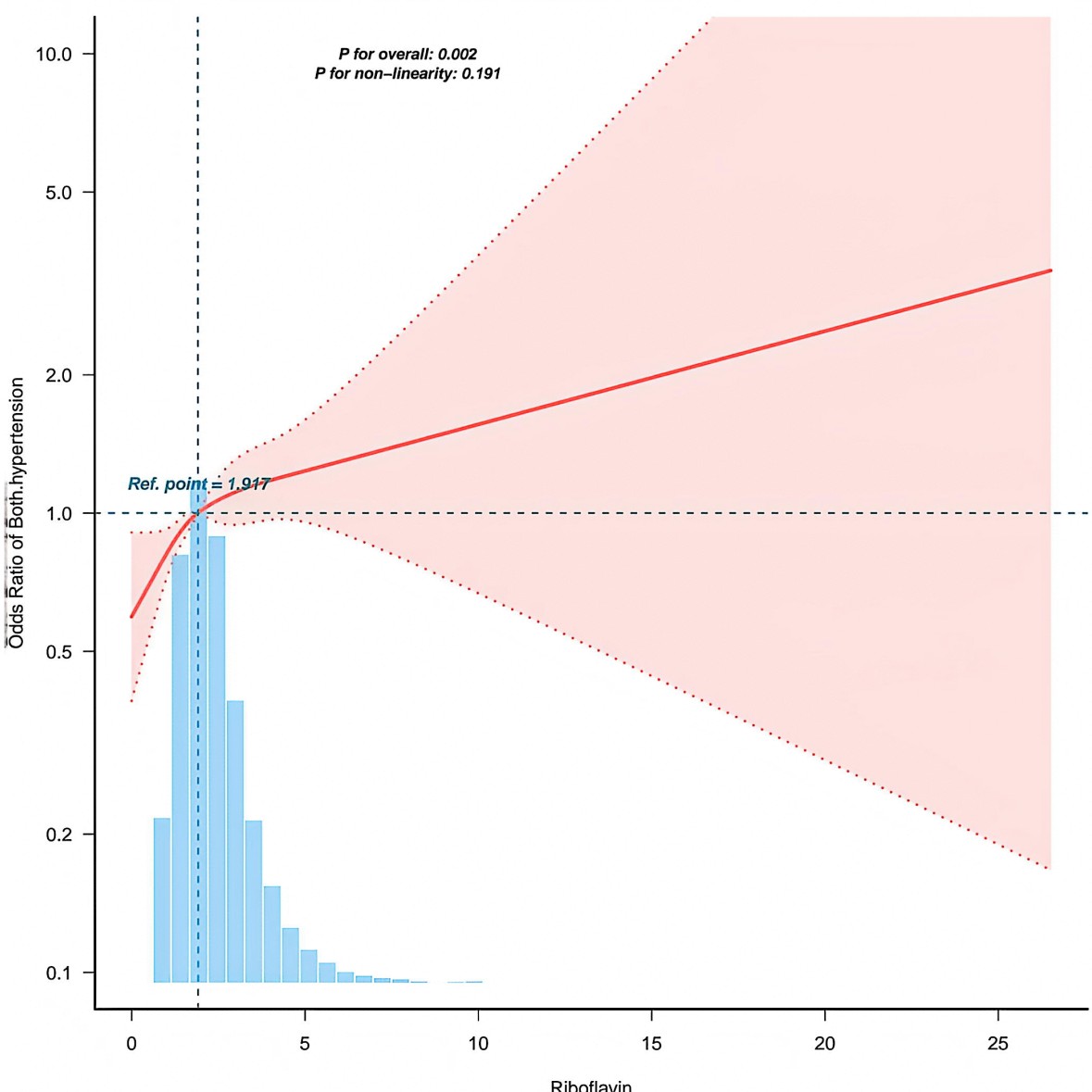

**Fig 2. Nonlinear relationship between riboflavin and the risk of systolic and diastolic hypertension.** The solid line indicates the pooled effect size (10.0), with shaded areas representing the 95% confidence interval. The overall effect test yielded P = 0.002; the non-linearity test yielded P = 0.191. The x-axis represents exposure dose (mg/day), while the y-axis represents the logarithm of the odds ratio.

non-linearity test yields P = 0.191, indicating no significant deviation from linearity. The risk of SDH exhibited a modest upward trend with increasing riboflavin levels, particularly pronounced in higher intake brackets. The reference point was set at the median intake level. Exploratory analysis using a restricted cubic spline model suggested a potential non-linear relationship. The overall association was significant (P = 0.002), though the test for non-linearity was not (P = 0.191). Visual inspection of the curve suggested a potential threshold effect around 6 mg/day. These findings suggest a potential dose-dependent association between riboflavin and SDH risk, though definitive confirmation of a non-linear relationship remains inconclusive. Table 5 reveals the threshold effect of riboflavin on systolic-diastolic hypertension. Segmented regression analysis identified a turning point at 6.046 mg. Below this threshold, each 1 mg increase in intake was associated with a 14% higher risk of disease (odds ratio OR=1.143, 95% confidence interval CI = 1.053–1.24, P = 0.0014); Beyond 6.046 mg, the risk growth curve markedly flattened (OR=0.917, P = 0.913). Neither the likelihood ratio test nor the nonlinearity test yielded statistical significance (P > 0.05), confirming this association is adequately described by a single threshold rather than a complex curve.

### Forest plot subgroup analysis

Fig 3 presents a forest plot evaluating the association between dietary thiamine intake and isolated systolic hypertension across subgroups. Gender (interaction p-value = 0.048) and smoking status (interaction p-value = 0.017) both demonstrated significant effect modification, with stronger positive correlations observed among males (odds ratio 1.22, 95% CI 1.14–1.30) and non-smokers (OR 1.24, 95% CI 1.16–1.32). No significant interactions were observed for household income, history of diabetes, cardiovascular disease history, BMI, ethnicity, or educational attainment (all interaction p-values > 0.05). Overall, findings suggest that the relationship between thiamine intake and hypertension risk may vary by sex and smoking behaviour, while remaining consistent across other sociodemographic and clinical factors. Fig 4 presents the results of the forest plot analysis, revealing the association between dietary riboflavin intake and isolated systolic hypertension across subgroups. Smoking status demonstrated a statistically significant interaction (p-value for interaction = 0.005), with non-smokers exhibiting a markedly stronger association (odds ratio 1.21, 95% confidence interval 1.15–1.28) than smokers. No significant effect modification was detected for gender, household income, history of diabetes, history of cardiovascular disease, BMI, ethnicity, or educational attainment (all interaction p-values > 0.05). Collectively, these findings indicate that the positive association between riboflavin intake and hypertension is more pronounced among non-smokers, while remaining consistent across other demographic and clinical strata. Fig 5 presents a forest plot analysing the association between dietary niacin intake and isolated systolic hypertension across different subgroups. Significant effect modification was observed for smoking status (interaction p-value = 0.03) and BMI classification (interaction p-value < 0.001). This association was more pronounced among non-smokers (OR=1.02, 95% CI 1.01–1.02) and overweight individuals (BMI = 2; OR=1.02, 95% CI 1.02–1.03). No significant interactions were observed for gender, household income, history of diabetes, history of cardiovascular disease, ethnicity, or educational attainment (all interaction p-values > 0.05). These findings suggest that the relationship between niacin and hypertension may be influenced by smoking status and body weight.

**Table 5. Analysis of the threshold effect of riboflavin intake in systolic and diastolic hypertension.**

| Riboflavin Intake | Adjusted Model | |
|---|---|---|
| | OR (95% CI) | P value |
| <6.046 | 1.143 (1.053~1.24) | 0.0014 |
| ≥6.046 | 0.917 (0.192~4.367) | 0.913 |
| Log-likelihood ratio test | – | 0.954 |

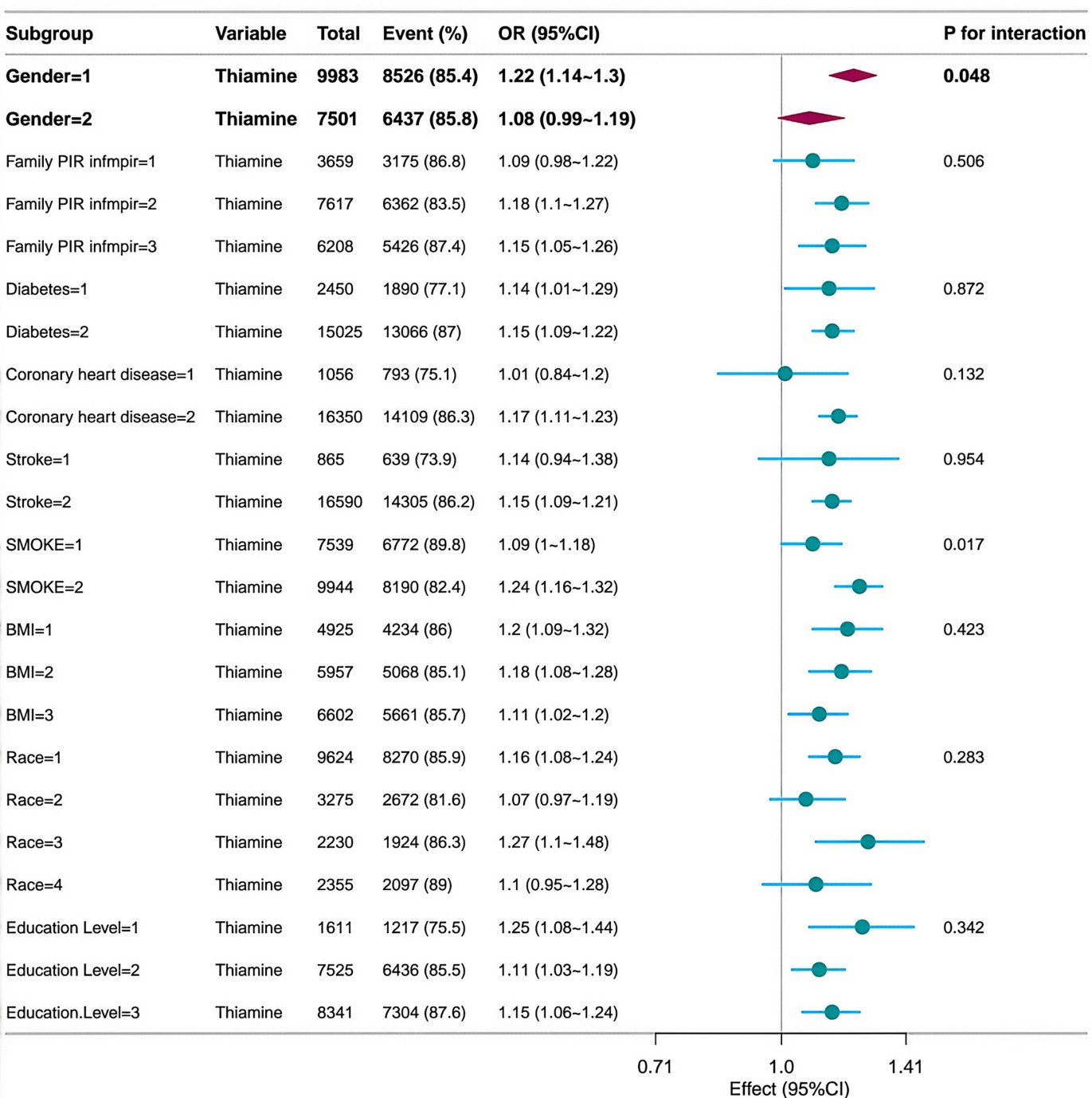

**Fig 3. Correlation between dietary thiamine intake and isolated systolic hypertension based on general characteristics.** B1 denotes dietary thiamine intake; BMI denotes body mass index (kg/m²); P for interaction values are shown; An investigation of the weighted dose-response association between dietary thiamine intake and isolated systolic hypertension across subgroups. Trend P-values were calculated via logistic regression, employing quartile medians as continuous variables; interaction P-values originated from likelihood ratio tests comparing models with and without interaction terms.

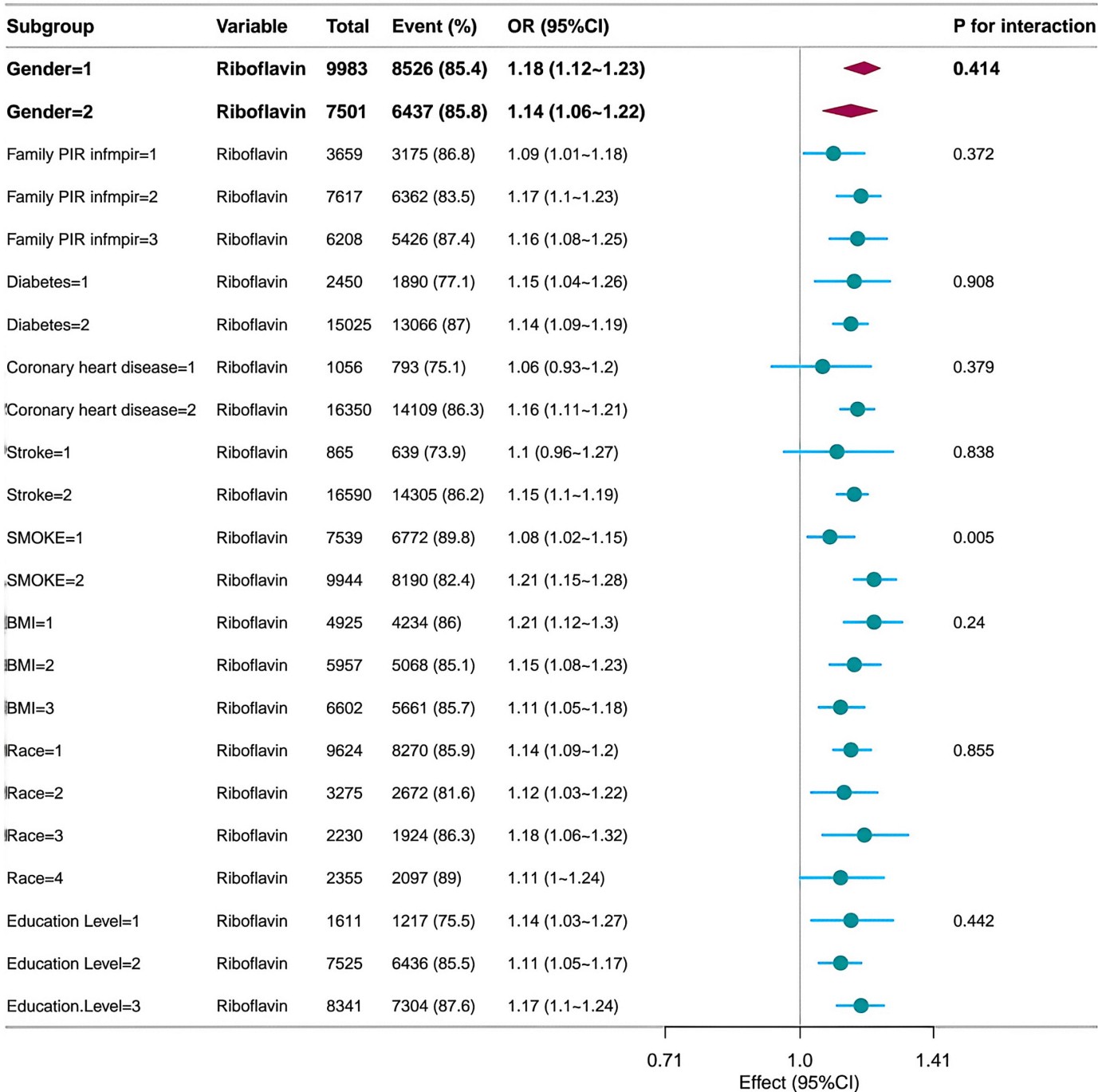

**Fig 4. Correlation between riboflavin intake and isolated systolic hypertension according to general characteristics.** B2 denotes dietary riboflavin intake; BMI denotes body mass index (kg/m²); P for interaction values are shown; Survey-weighted dose–response association between dietary riboflavin intake and isolated systolic hypertension across subgroups. P-trend were calculated by logistic regression with quartile medians as continuous variables; interaction P-values were derived from likelihood-ratio tests comparing models with and without interaction terms.

| Subgroup | Variable | Total | Event (%) | OR (95%CI) | | P for interaction |
|----------|----------|-------|-----------|------------|---|-------------------|
| **Gender=1** | **Niacin** | **9983** | **8526 (85.4)** | **1.02 (1.01~1.02)** | | **0.218** |
| **Gender=2** | **Niacin** | **7501** | **6437 (85.8)** | **1.01 (1~1.02)** | | |
| Family PIR infmpir=1 | Niacin | 3659 | 3175 (86.8) | 1.01 (1.01~1.02) | | 0.244 |
| Family PIR infmpir=2 | Niacin | 7617 | 6362 (83.5) | 1.02 (1.01~1.02) | | |
| Family PIR infmpir=3 | Niacin | 6208 | 5426 (87.4) | 1.01 (1~1.01) | | |
| Diabetes=1 | Niacin | 2450 | 1890 (77.1) | 1.01 (1~1.02) | | 0.413 |
| Diabetes=2 | Niacin | 15025 | 13066 (87) | 1.01 (1.01~1.02) | | |
| Coronary heart disease=1 | Niacin | 1056 | 793 (75.1) | 1.01 (0.99~1.02) | | 0.408 |
| Coronary heart disease=2 | Niacin | 16350 | 14109 (86.3) | 1.01 (1.01~1.02) | | |
| Stroke=1 | Niacin | 865 | 639 (73.9) | 1.02 (1~1.03) | | 0.385 |
| Stroke=2 | Niacin | 16590 | 14305 (86.2) | 1.01 (1.01~1.02) | | |
| SMOKE=1 | Niacin | 7539 | 6772 (89.8) | 1.01 (1~1.01) | | 0.03 |
| SMOKE=2 | Niacin | 9944 | 8190 (82.4) | 1.02 (1.01~1.02) | | |
| BMI=1 | Niacin | 4925 | 4234 (86) | 1.01 (1~1.02) | | <0.001 |
| BMI=2 | Niacin | 5957 | 5068 (85.1) | 1.02 (1.02~1.03) | | |
| BMI=3 | Niacin | 6602 | 5661 (85.7) | 1.01 (1~1.01) | | |
| Race=1 | Niacin | 9624 | 8270 (85.9) | 1.01 (1.01~1.01) | | 0.113 |
| Race=2 | Niacin | 3275 | 2672 (81.6) | 1.01 (1~1.02) | | |
| Race=3 | Niacin | 2230 | 1924 (86.3) | 1.02 (1.01~1.03) | | |
| Race=4 | Niacin | 2355 | 2097 (89) | 1.02 (1.01~1.03) | | |
| Education Level=1 | Niacin | 1611 | 1217 (75.5) | 1.01 (1~1.02) | | 0.838 |
| Education Level=2 | Niacin | 7525 | 6436 (85.5) | 1.01 (1.01~1.02) | | |
| Education.Level=3 | Niacin | 8341 | 7304 (87.6) | 1.01 (1.01~1.02) | | |

0.71                    1.0
Effect (95%CI)

**Fig 5. Correlation between dietary niacin intake and isolated systolic hypertension based on general characteristics.** B3 denotes dietary niacin intake; BMI denotes body mass index (kg/m$^2$); P for interaction values are shown; Dietary niacin intake and weighted dose-response associations with isolated systolic hypertension across subgroups. Trend P-values were calculated via logistic regression using quartile medians as continuous variables; interaction P-values derived from likelihood ratio tests comparing models with and without interaction terms.

## Discussion

This large-scale, nationally representative cross-sectional study represents the first comprehensive analysis of dietary intake of three key B vitamins (thiamine, riboflavin, and niacin) and their association with distinct hypertension subtypes. Key findings reveal a significant, independent, and dose-dependent positive association between higher riboflavin intake and increased risk of systolic-diastolic hypertension (SDH), contrasting markedly with most existing studies focusing on riboflavin's antihypertensive potential. Following comprehensive multivariable adjustment, the initial positive correlation between B vitamins and isolated systolic hypertension (ISH) largely dissipated, highlighting the critical role of confounding factors. No consistent association was observed for isolated diastolic hypertension (IDH).

The positive correlation observed between riboflavin and SDH appears to contradict substantial evidence indicating that riboflavin supplementation possesses antihypertensive effects. Randomised controlled trials and targeted studies consistently demonstrate that riboflavin supplementation significantly reduces systolic blood pressure, particularly in individuals carrying the MTHFR 677TT genotype [12,17-18–]. Several explanations may reconcile this apparent contradiction. Foremost and most crucially, the aforementioned intervention evidence applies exclusively to a specific genetic subpopulation characterised by impaired folate metabolism and elevated homocysteine levels, where riboflavin functions as a key cofactor for MTHFR enzyme activity within this metabolic pathway [19]. The present study assessed habitual intake in an unscreened general population, reflecting aggregate effects at the population level. This may differ from therapeutic correction of metabolic defects in high-risk subpopulations. Secondly, in observational epidemiology, high dietary riboflavin intake may serve as a marker for other dietary patterns (such as high consumption of fortified cereals, dairy products, and meat) or unmeasured lifestyle factors associated with SDH risk, rather than directly indicating causal adverse effects. Thirdly, we hypothesise the existence of a non-linear or bidirectional relationship. Whilst riboflavin can correct endothelial dysfunction and reduce homocysteine levels in deficient or high-risk states [12,20], excessive intake in nutritionally adequate populations may disrupt cellular redox balance or other metabolic pathways. This nutrient overload hypothesis is supported by research demonstrating complex dose-dependent effects of micronutrients [21]. The threshold effect we observed—where SDH risk plateaued after riboflavin intake increased to approximately 6 mg/day—provides preliminary epidemiological support for this non-linear model. Moreover, riboflavin exhibited a stronger positive correlation with ISH risk among non-smokers (Fig 4), indicating that lifestyle factors significantly modulate dietary effects. This suggests smoking may overwhelm or alter the metabolic pathways through which riboflavin influences vascular tone.

After multivariate adjustment, the disappearance of the significant association between B vitamins and ISH aligns with the understanding that ISH primarily serves as a haemodynamic phenotype of arterial stiffness, significantly influenced by age and long-term vascular ageing [1,4]. Numerous cross-sectional studies reporting associations between nutrients and hypertension struggle to fully account for these potent confounding factors [15]. This study underscores the critical importance of comprehensive variable adjustment in nutritional epidemiology, particularly concerning age-related diseases. As shown in Table 1, the initial positive signal may stem from the ISH group being older and exhibiting poorer overall nutritional status, rather than vitamins themselves exerting a direct pathogenic effect.

Thiamine, also known as vitamin B1, is an essential water-soluble micronutrient for the human body [22]. Thiamine has a relatively short retention time in the body, lasting only approximately one to three weeks, and cannot be synthesised by the human body itself [23]. Thiamine is primarily obtained through dietary intake or supplementation. Key food sources include pulses, whole grains, nuts, and coarse cereals. Furthermore, thiamine serves as an essential coenzyme in glucose metabolism. In contemporary diets characterised by high-calorie food consumption and reduced intake of coarse grains, the demand for thiamine has increased, heightening susceptibility to deficiency [24]. Extensive literature demonstrates that thiamine deficiency elevates the risk of multiple systemic diseases [25]. Thiamine may improve or reverse angina pectoris, myocardial infarction, cardiovascular disease, diabetes mellitus, dyslipidaemia, obesity, and psychiatric disorders [11,26]. Consuming a measured amount of thiamine may improve cardiac function [27]. Reduce systemic vascular resistance [28] and haemodynamic characteristics [29]. Moreover, it represents an effective and low-cost intervention.

Research indicates that lower limb arterial smooth muscle cells play a pivotal role in maintaining vascular elasticity and preventing atherosclerosis. Thiamine exerts significant protective effects on smooth muscle cells through insulin and glucose-mediated pathways. However, large-scale population studies examining the association between thiamine intake and hypertension subgroups or pulse pressure remain scarce. This research addresses that gap.

Dietary niacin intake showed no independent association with any subtype of hypertension, whilst quartile analysis revealed only a weak non-linear signal for thiamine and SDH. A Korean cross-sectional study suggested a negative association between thiamine intake and hypertension risk [11], whereas large Chinese cohort studies described a J-shaped relationship between niacin intake and new-onset hypertension, with the lowest risk observed at moderate intake levels [13,30]. Discrepancies with the present findings may stem from multiple factors: differences in study populations (Asian versus American), variations in dietary source composition, the present analysis's focus on hypertension subtypes, and the combined analysis of food and supplement sources potentially obscuring effects from specific sources. Supplementary analyses (S1 Table) indeed revealed no consistent signal for either vitamin across all sources, further corroborating the overall lack of significant association. This indicates that, within the intake ranges observed in the US population, dietary thiamine and niacin are not major independent determinants of risk for hypertensive subtypes.

The mechanism underlying the potential adverse association between high riboflavin intake and SDH remains speculative but warrants investigation. SDH frequently represents an earlier, more active stage of hypertension, involving sympathetic activation and endothelial injury. Whilst riboflavin is crucial for antioxidant activity (e.g., via the glutathione reductase pathway), excessive intake may cause it or its metabolites to act as pro-oxidants under specific cellular conditions, or interfere with the metabolism of other vasoactive nutrients. Alternatively, high-dose riboflavin may adversely affect sodium balance or vascular smooth muscle cell proliferation via flavoprotein-dependent pathways. These mechanistic hypotheses require explicit validation through experimental models.

The following limitations warrant consideration: firstly, the cross-sectional design precludes establishing causality. Reverse causality may exist if hypertensive patients alter dietary habits or supplement usage; secondly, dietary assessment relied on the 24-hour dietary recall method, which is susceptible to measurement error and may inaccurately reflect habitual intake; Thirdly, despite adjusting for numerous potential confounders, residual confounding may persist. Fourthly, the limited sample size for isolated diastolic hypertension reduced the power for detecting associations.

Despite these limitations, the present study possesses significant strengths, including: a large, nationally representative sample; standardised measurement protocols; comprehensive assessment of dietary and supplemental vitamin intake; and careful consideration of hypertension subtypes. The use of survey-weighted analyses enhances the generalisability of findings to the US adult population. From a clinical perspective, our findings suggest that recommendations for B-vitamin supplementation in blood pressure management lack sufficient evidence. Maintaining balanced dietary intake remains the prudent choice. For individuals with systolic-predominant hypertension or those at high risk, monitoring riboflavin intake may be warranted, though prospective studies are required to validate associations and determine optimal intake ranges.

Future research should address several key questions: prospective cohort studies could clarify causal sequences and reduce reverse causality bias; Randomised controlled trials comparing different riboflavin intake levels could establish optimal ranges; Mechanistic studies should investigate riboflavin's effects on vascular function at varying intake levels; Finally, gene-nutrient interaction research (particularly regarding methionine synthase reductase polymorphisms) could help identify subgroups most likely to benefit from riboflavin interventions.

## Shortcomings and outlook

In conclusion, this study provides novel epidemiological evidence that dietary riboflavin intake is positively associated with ISH and SDH in the general US adult population, with a potential intake threshold for SDH risk. Thiamine showed a weaker association with SDH, while niacin and all three vitamins showed no association with IDH. These findings highlight the potential for hypertension subtypes to have distinct nutritional etiologies. They suggest that blanket recommendations

for B vitamin supplementation for blood pressure management are not supported by this evidence. Instead, a focus on maintaining a balanced diet appears prudent. Future research, particularly prospective cohort studies and well-designed randomized trials in general populations, is necessary to confirm these associations, elucidate the underlying mechanisms, and determine whether specific recommendations for riboflavin intake should be considered in public health strategies for hypertension prevention.

## Conclusion

In summary, this study reveals distinct epidemiological and nutritional associations across different hypertension subtypes. A key finding is the significant independent positive correlation between higher dietary riboflavin intake and increased risk of combined systolic-diastolic hypertension, exhibiting a dose-response relationship. Conversely, the initial positive association between B vitamins and isolated systolic hypertension failed to persist after comprehensive adjustment, suggesting this link is primarily confounded by factors such as age and comorbidities.

These results underscore the importance of distinguishing hypertension subtypes in aetiological research. The strong association between riboflavin and SDH warrants further investigation into potential biological mechanisms. The effect-modifying role of smoking status on several associations suggests that lifestyle factors may interact with dietary components to influence hypertension risk. The potential threshold effect of riboflavin (though requiring further validation) implies an optimal intake range beyond which risk does not increase proportionally.

The final conclusion is that while dietary factors are important, their relationship with hypertension is complex and subtype-specific. This analysis underscores that associations observed in unadjusted models do not necessarily represent independent causation. Future research should prioritise longitudinal designs and mechanistic studies to clarify whether these associations—particularly that between riboflavin and SDH—reflect causation, residual confounding, or markers of other dietary or lifestyle patterns.

## Supporting information

**S1 Table. Associations of vitamin intake from food and supplement sources with hypertension subtypes.** ISH: Isolated systolic hypertension; IDH: Isolated diastolic hypertension; SDH: Systolic-diastolic hypertension.
(DOCX)

**S2 Table. Fully anonymized version.**
(XLSX)

## Acknowledgments

The authors acknowledge Jie Liu of the Department of Vascular and Endovascular Surgery, Chinese PLA General Hospital for his contribution to the statistical support, study design consultations, and comments regarding the manuscript. We would like to thank all participants in this study.

## Author contributions

**Conceptualization:** Siqi Ma.

**Data curation:** Qu Jin, Keying Yu, Hezeng Dong, Yanmei Lu.

**Methodology:** Siqi Ma.

**Resources:** Liping Chang.

**Software:** Qu Jin, Keying Yu, Hezeng Dong, Yanmei Lu.

**Supervision:** Liping Chang.

**Writing – original draft:** Siqi Ma.

**Writing – review & editing:** Liping Chang.

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
