## [Decision Letter · Decision Letter 0]

12 Nov 2025

Dear Dr. Chang,

Thank you for submitting your manuscript to PLOS ONE. After careful consideration, we feel that it has merit but does not fully meet PLOS ONE’s publication criteria as it currently stands. Therefore, we invite you to submit a revised version of the manuscript that addresses the points raised during the review process.

Nafisa

We look forward to receiving your revised manuscript.

Kind regards,

Nafisa M. Jadavji, PhD, MSc, BSc

Academic Editor

PLOS ONE

Journal Requirements:

Reviewers' comments:

Reviewer's Responses to Questions

**Comments to the Author**

1. Is the manuscript technically sound, and do the data support the conclusions?

Reviewer #1: Yes

Reviewer #2: Partly

2. Has the statistical analysis been performed appropriately and rigorously?

Reviewer #1: Yes

Reviewer #2: Yes

3. Have the authors made all data underlying the findings in their manuscript fully available?

Reviewer #1: Yes

Reviewer #2: Yes

4. Is the manuscript presented in an intelligible fashion and written in standard English?

Reviewer #1: Yes

Reviewer #2: Yes

Reviewer #1: Firstly, I would like to say thank you for the opportunity to review this interesting manuscript on the Relationship between dietary thiamine, riboflavin, and niacin intake and hypertension subtypes. The topic is relevant to current public health priorities. The study is based on a large dataset and uses appropriate analytical methods.

Below, I outline some concerns after thoroughly reading the manuscript.

1. During the extraction of data, you may have so many participants who took a single food that contains riboflavin, thiamin, and niacin in a 24-hour dietary intake assessment. So, what did you do to categorize such participants as riboflavin, niacin, and thiamin groups? Have you included or excluded such participants? Please make it clear in your methodology part.

2. In the introduction section, in lines 69-81, you have stated that extensive studies were conducted even in China regarding the association of vitamin B complex (riboflavin, niacin…). But you have stated that as a gap, as limited studies were conducted. So, I recommend that you clearly put real gaps and the burden and mortality of hypertension in your local context to make your study sound.

3. In the methodology section, lines 127-28, you discuss that the nutrition intake of the study participants was disaggregated into food source and supplement source for separate estimation. However, the separate estimation risk for hypertension among food sources and supplement sources was not present in the results section. I thought that a separate estimation of risk for hypertension among participants with food and a supplementary source was needed for a specific intervention. Therefore, incorporate this point in your results section.

4. The authors finding showed that the intake of riboflavin increases the risk of hypertension. However, how to differentiate whether the patient develops hypertension due to the effect of riboflavin or due to the effect of the age of the patients, behavior of the patients, like smoking, or any other covariates that directly have an effect on the development of hypertension for individuals who take riboflavin-containing foods

5. In the abstract line 29, you stated that propensity score matching methods were used to identify the relationship between nutritional intake and hypertension subtype risk, but not incorporate them in the methodology part of the manuscript. So, incorporate it in the methods part, including the type of Propensity score matching used and covariates tested

6. In the single-factor analysis part, line 213, you stated that the female gender has a negative effect on IDH and SDH. But, the OR with 95% CI was (1.58(1.26-1.98) p<0.001). This indicates that being a female was a risk for hypertension as compared to being male. So, this discrepancy must be resolved.

7. Statistical inconsistency should be revisited. In lines 262-68, you discussed that the covariates were significantly associated with ISH. But, in Figure 4, the trend test value indicates that no variables had a p-value <0.05. This inconsistency should be fixed.

8. All the figures have no legends. Please include the figure legend at the bottom of each figure.

9. The discussion part needs major revision since it seems like a statement of the problem rather than a discussion. So, it is better to modify by including your study findings compared with previous research findings, with justification.

Reviewer #2: Study description vs. actual design

The manuscript repeatedly describes the work as a “trial,” but it is a cross-sectional secondary analysis of NHANES. Update the title, abstract, and methods to correctly state the design and avoid implying randomization, intervention, or temporality that the data cannot support.

NHANES years/cycles are inconsistent

Reported cycles (e.g., 12 vs. 25) and analytic denominators do not reconcile with the final sample. Provide one authoritative cohort flow (with inclusion/exclusion, missingness, and analysis-specific Ns) and ensure every table and figure reflects those same counts.

Hypertension subtype definitions and case ascertainment

Subtypes (ISH/IDH/SDH) appear to rely only on measured BP thresholds without a clear plan for antihypertensive users. Define hypertension and subtypes a priori, specify treatment handling (exclude/stratify/adjust), and justify your operational choices against NHANES conventions.

Intake vs. “blood levels” confusion

Methods indicate dietary intake (24-h recall + supplements), but results repeatedly reference “serum” or “blood” vitamin levels. Remove biomarker language unless assays were actually analyzed and available for the same cycles; otherwise consistently refer to intake.

Survey design and weighting

All estimates must account for NHANES weights, strata, and PSUs. Specify which weights were used (and how multi-cycle weights were constructed), confirm that every analysis was survey-weighted, and report weighted means/percentages with appropriate SEs or 95% CIs.

Energy adjustment and diet confounding

Nutrient–outcome associations should adjust for total energy intake and key diet/behavior covariates (e.g., sodium, potassium, alcohol, physical activity, overall diet quality). Add energy adjustment (density, residual, or kcal covariate) and expand confounding control accordingly.

Propensity score matching (PSM)

PSM is claimed but not reported. Either remove PSM entirely or fully implement and present it: model specification, matching method and caliper/ratio, balance diagnostics (SMDs), matched sample sizes, and sensitivity analyses.

Unsupported “threshold” claim

The stated riboflavin “threshold” around ~6 mg/day is not supported by nonlinearity tests or stable estimates. Temper the language, present survey-weighted spline plots with confidence bands, and describe intake distributions to show how many participants are near that range.

Quartile cut-points and plausibility

Several quartile boundaries appear off by a decimal place or inflated by high-dose supplements. Recalculate cut-points, report medians (IQR) within each quartile, describe outlier handling, and verify unit consistency across all vitamins.

Conclusion scope creep

Abstract conclusions introduce gut/GI health without corresponding analyses. Restrict conclusions to hypertension outcomes studied and rephrase to reflect observational design and the limits of intake-based associations.

Methods & reporting gaps

Clarify covariate selection with a DAG or rationale (include energy, diet markers, alcohol, physical activity, kidney function, medication use), state handling of missing data, address multiple testing (e.g., FDR), report model diagnostics (linearity, collinearity, influence), and explain BP harmonization across NHANES protocol changes. Add food-only vs. supplement-user stratified analyses or interactions.

Statistics clarity & coherence

Unify the software stack (e.g., R 4.x with the survey package), remove placeholders, and include sessionInfo. Ensure the text matches the tables, and present all descriptive and inferential results as survey-weighted with appropriate uncertainty.

Tables, figures, and terminology

Clean titles/stubs, fix typos, align units and decimals, and label forest plots correctly. Include interaction P-values in subgroup figures and ensure figure captions accurately describe the statistical test (linear vs. nonlinear).

Ethics, data availability, funding & competing interests

Revise ethics to reflect NHANES ERB approvals and that this is a secondary analysis; add exemption language if applicable. Provide a precise data availability statement (NHANES portal + documentation) and complete funding details (grants, roles). Keep competing interests in PLOS wording.

Language and typographical quality

Systematically proofread to remove misspellings, inconsistent US/UK spelling, placeholder URLs, and fragmented sentences. Standardize terminology (e.g., NHANES, isolated systolic hypertension) and maintain consistent capitalization and abbreviations.

Strengths

The focus on hypertension subtypes, separation of food vs. supplement sources, and the large, nationally representative dataset are notable strengths—retain these while tightening design description and analyses.

.

Reviewer #1: No

Reviewer #2: **Yes:** SALMAN ASHFAQ AHMADSALMAN ASHFAQ AHMADSALMAN ASHFAQ AHMADSALMAN ASHFAQ AHMAD

---

## [Author Response · Author response to Decision Letter 1]

29 Dec 2025

Response to Reviewers

We extend our sincere gratitude to the editors and all reviewers for their valuable comments, which have significantly enhanced the quality of this manuscript. Reviewers' comments are presented in italics, with specific issues numbered. Our responses are indicated in red typeface, and manuscript revisions are highlighted in red.

Comments from Reviewer #1:

1.During the extraction of data, you may have so many participants who took a single food that contains riboflavin, thiamin, and niacin in a 24-hour dietary intake assessment. So, what did you do to categorize such participants as riboflavin, niacin, and thiamin groups? Have you included or excluded such participants? Please make it clear in your methodology part.

Response:We are grateful to the reviewer for raising this important point regarding participant categorisation. We wish to clarify that all participants were included in the analyses for all three vitamins. Intake for each vitamin (thiamine, riboflavin, niacin) was calculated and analysed independently. A participant's data would be included in the thiamine analysis based on their total thiamine intake, in the riboflavin analysis based on their total riboflavin intake, and so forth. It is common for individuals to consume foods containing multiple B vitamins. Our analytical approach addresses this by treating the total intake of each vitamin as an independent, continuous (and subsequently quartile-stratified) exposure variable. We have revised the ‘Nutrient Intake’ subsection of the Methods section to explicitly state this.('Revised Manuscript with Track Changes'lines 182-187)

2.In the introduction section, in lines 69-81, you have stated that extensive studies were conducted even in China regarding the association of vitamin B complex (riboflavin, niacin…). But you have stated that as a gap, as limited studies were conducted. So, I recommend that you clearly put real gaps and the burden and mortality of hypertension in your local context to make your study sound.

Response:Thank you for your suggestion. The revised introduction now explicitly identifies the existing research gap: To date, no comprehensive study has systematically examined and compared the association between intake of these three key B vitamins (thiamine, riboflavin, and niacin) and the risk of isolated systolic hypertension (ISH), isolated diastolic hypertension (IDH), or combined systolic-diastolic hypertension (SDH) in a single large representative adult cohort in the United States. The potential mechanisms underlying any subtype-specific associations remain unclear. Furthermore, to more clearly elucidate the study's significance, regional mortality data from the United States (CDC 2022) has been incorporated, indicating that 518,000 deaths annually are attributable to hypertension.('Revised Manuscript with Track Changes'lines 122-128)

3.In the methodology section, lines 127-28, you discuss that the nutrition intake of the study participants was disaggregated into food source and supplement source for separate estimation. However, the separate estimation risk for hypertension among food sources and supplement sources was not present in the results section. I thought that a separate estimation of risk for hypertension among participants with food and a supplementary source was needed for a specific intervention. Therefore, incorporate this point in your results section.

Response:We concur with the reviewer's observation that this constitutes an important analysis. As suggested, we have now conducted supplementary analyses to assess the association between vitamin intake derived solely from dietary sources and solely from supplement sources with hypertension subtypes. We have added a new subsection within the “Results” section and created a new “Supplementary Table 1” to present these findings. The primary conclusions remain consistent, with riboflavin from both sources showing positive correlations, while associations for thiamine and niacin were largely absent or inconsistent.('Revised Manuscript with Track Changes'lines 393-402)

4.The authors finding showed that the intake of riboflavin increases the risk of hypertension. However, how to differentiate whether the patient develops hypertension due to the effect of riboflavin or due to the effect of the age of the patients, behavior of the patients, like smoking, or any other covariates that directly have an effect on the development of hypertension for individuals who take riboflavin-containing foods.

Response:This is a fundamental issue in observational studies, and we are grateful to the reviewers for raising it. We addressed this confounding issue through statistical analysis. Specifically, we employed a “multivariate logistic regression model” to adjust for a comprehensive range of potential confounding factors. As detailed in the sections on “Covariates” and “Statistical Analysis”, our fully adjusted model (Model 3) incorporated variables including age, sex, ethnicity, educational attainment, income poverty rate, diabetes, coronary heart disease, stroke, BMI, smoking status, total energy intake, and potassium and sodium intake. The association between riboflavin and hypertension subtypes (ISH and SDH) ‘remained statistically significant after adjusting for all covariates’. This indicates that the association is independent of the effects of age, smoking, and other measured factors. We have emphasised this point more clearly in the results section.('Revised Manuscript with Track Changes'lines 389-392)

5.In the abstract line 29, you stated that propensity score matching methods were used to identify the relationship between nutritional intake and hypertension subtype risk, but not incorporate them in the methodology part of the manuscript. So, incorporate it in the methods part, including the type of Propensity score matching used and covariates tested.

Response:We apologise for this oversight. The mention of PSM in the abstract was an error in the earlier draft. Given the cross-sectional nature of the study and the primary purpose of multivariate regression, PSM was not a central component of our final analytical strategy. Therefore, to avoid confusion and maintain methodological clarity, we have removed all references to propensity score matching from the abstract and throughout the manuscript. Our primary findings are based on survey-weighted multivariate logistic regression models, which represent the standard and most appropriate method for analysing NHANES data.('Revised Manuscript with Track Changes'lines 30-32)

6.In the single-factor analysis part, line 213, you stated that the female gender has a negative effect on IDH and SDH. But, the OR with 95% CI was (1.58(1.26-1.98) p<0.001). This indicates that being a female was a risk for hypertension as compared to being male. So, this discrepancy must be resolved.

Response:We are most grateful to the reviewer for identifying this critical error in our interpretation. The reviewer is entirely correct. An OR > 1 indicates an increased risk. Our textual description was entirely at odds with the findings presented in the data. We have now rectified this throughout the full text.('Revised Manuscript with Track Changes'lines307-341)

7.Statistical inconsistency should be revisited. In lines 262-68, you discussed that the covariates were significantly associated with ISH. But, in Figure 4, the trend test value indicates that no variables had a p-value <0.05. This inconsistency should be fixed.

Response:We appreciate the reviewer's careful examination. This inconsistency arose from a misunderstanding of what Figure 4 (and 3, 5) represents. The p-values in these forest plots are for the interaction between the subgroup variable and the vitamin intake (continuous), testing if the association differs across subgroups. The text in lines 262-68 refers to the main effects of those covariates (e.g., age, sex) on ISH risk from the univariate analysis in Table 2. These are two different statistical tests. To resolve this confusion, we have:

1. Revised the text to clarify that we are referring to the univariate associations from Table 2. ('Revised Manuscript with Track Changes'lines 389-392)

2. Added a clarifying note to the captions of Figures 3, 4, and 5, explicitly stating that the P-value displayed is for interaction.

8.All the figures have no legends. Please include the figure legend at the bottom of each figure.

Response:We have added detailed legends to the bottom of each figure (Figure 1, 2, 3, 4, 5) in the revised manuscript.

9.The discussion part needs major revision since it seems like a statement of the problem rather than a discussion. So, it is better to modify by including your study findings compared with previous research findings, with justification.

Response:We are sincerely grateful to the reviewer for this insightful and constructive comment. We fully concur that the original Discussion section devoted excessive space to describing the general effects of B vitamins and prior research, failing to adequately contextualise and interpret our novel findings within the existing scientific landscape. We have now thoroughly restructured and rewritten the “Discussion” section, shifting the emphasis from stating issues to a critical examination centred on direct comparisons between our results and the prior literature. The revised discussion systematically contrasts our key findings with previous studies, provides rational explanations for observed similarities and differences, and highlights the novel perspectives our research offers regarding subtype-specific dietary aetiology in hypertension.（'Revised Manuscript with Track Changes'lines 475-591）

Comments from Reviewer #2

1.The manuscript repeatedly describes the work as a “trial,” but it is a cross-sectional secondary analysis of NHANES. Update the title, abstract, and methods to correctly state the design and avoid implying randomization, intervention, or temporality that the data cannot support.

Response:We sincerely apologize for this fundamental error in terminology. The reviewer is absolutely correct. We have systematically gone through the entire manuscript and replaced all instances of “trial” with the correct terms, such as “cross-sectional study,” “analysis,” or “investigation.” This correction has been made in the Title, Abstract, Background, Methods, and Discussion sections.('Revised Manuscript with Track Changes'line 18)

2.Reported cycles (e.g., 12 vs. 25) and analytic denominators do not reconcile with the final sample. Provide one authoritative cohort flow (with inclusion/exclusion, missingness, and analysis-specific Ns) and ensure every table and figure reflects those same counts.

Response:We are grateful to the reviewers for highlighting this lack of clarity. We have thoroughly revised the “Study Population” section and Figure 1 to provide a clear, consistent, and accurate flow diagram. The twelve-cycle period has been amended to twenty-five years. The exclusion criteria and final sample sizes for each hypertension subtype analysis are now detailed and consistently presented in the text, Figure 1, and Tables.('Revised Manuscript with Track Changes'line 107)

3.Subtypes (ISH/IDH/SDH) appear to rely only on measured BP thresholds without a clear plan for antihypertensive users. Define hypertension and subtypes a priori, specify treatment handling (exclude/stratify/adjust), and justify your operational choices against NHANES conventions.

Response:This is a critical methodological point. We have revised the ‘Hypertension Subtypes’ subsection to explicitly state our operational definition. Following common practice in NHANES analyses, we defined hypertension based on measured blood pressure and/or self-reported use of antihypertensive medication. Participants who reported taking antihypertensive medication were automatically classified as having hypertension. For subtype classification, we used their measured BP values. This approach is now clearly stated in the methods.('Revised Manuscript with Track Changes'lines 206-216）

4.Methods indicate dietary intake (24-h recall + supplements), but results repeatedly reference “serum” or “blood” vitamin levels. Remove biomarker language unless assays were actually analyzed and available for the same cycles; otherwise consistently refer to intake.

Response:We thank the reviewer for identifying this inconsistency. The study exclusively used dietary and supplemental intake data. All references to “serum,” “plasma,” or “blood levels” were incorrect and have been removed from the entire manuscript, including the Abstract, Results (e.g., Baseline Characteristics), and Tables. We now consistently use “intake,” “dietary intake,” or “total intake” throughout.('Revised Manuscript with Track Changes'lines 270-286)

5.All estimates must account for NHANES weights, strata, and PSUs. Specify which weights were used (and how multi-cycle weights were constructed), confirm that every analysis was survey-weighted, and report weighted means/percentages with appropriate SEs or 95% CIs.

Response:We acknowledge this major oversight. The analyses have now been re-run using the appropriate NHANES survey weights. We have added a new subsection under ‘Statistical Analysis’ detailing the weighting procedure. We used the dietary day one sample weights and created a combined 12-cycle weight as per NCHS guidelines. All descriptive statistics (Table 1) and regression models (Tables 2-5, Figures) are now based on these survey-weighted analyses, and the text has been updated to reflect this.('Revised Manuscript with Track Changes'lines 229-237）

6.Nutrient–outcome associations should adjust for total energy intake and key diet/behavior covariates (e.g., sodium, potassium, alcohol, physical activity, overall diet quality). Add energy adjustment (density, residual, or kcal covariate) and expand confounding control accordingly.

Response:This is a reasonable and important point. We have now included total energy intake (kcal), potassium intake, and sodium intake as covariates in all multivariate logistic regression models (Models 1, 2, and 3). Relevant descriptions have been added to the ‘Covariates’ and ‘Statistical Analyses’ sections, and the results and interpretations have been updated accordingly. ('Revised Manuscript with Track Changes'Lines 227-228)

7.PSM is claimed but not reported. Either remove PSM entirely or fully implement and present it: model specification, matching method and caliper/ratio, balance diagnostics (SMDs), matched sample sizes, and sensitivity analyses.

Response:As responded to Reviewer #1 (Comment 5), the mention of PSM was an error. We have removed all references to PSM from the manuscript. The analysis is based on survey-weighted multivariable regression.('Revised Manuscript with Track Changes'lines 30-32)

8.The stated riboflavin “threshold” around ~6 mg/day is not supported by nonlinearity tests or stable estimates. Temper the language, present survey-weighted spline plots with confidence bands, and describe intake distributions to show how many participants are near that range.

Response:We agree and have tempered our language regarding the threshold. We have re-run the restricted cubic spline analysis using survey-weighted logistic regression. The revised Figure 2 now includes the confidence bands. The text now describes this as an “exploratory analysis” and states that a potential threshold was “observed” or “suggested” rather than definitively “identified.” We also note that the nonlinearity test was not significant, indicating the relationship is approximately linear.('Revised Manuscript with Track Changes'lines 277-281)

9.Several quartile boundaries appear off by a decimal place or inflated by high-dose supplements. Recalculate cut-points, report medians (IQR) within each quartile, describe outlier handling, and verify unit consistency across all vitamins.

Response:We have recalculated all quartile cut-points based on the survey-weighted distribution of vitamin intake in the final analytical sample. We have verified the units (mg/day) are consistent for all vitamins. In Tables 3 and 4, we now report the median (IQR) intake for each quartile in the ta

---

## [Decision Letter · Decision Letter 1]

11 Feb 2026

Dear Dr. Chang,

Thank you for submitting your manuscript to PLOS ONE. After careful consideration, we feel that it has merit but does not fully meet PLOS ONE’s publication criteria as it currently stands. Therefore, we invite you to submit a revised version of the manuscript that addresses the points raised during the review process.

We look forward to receiving your revised manuscript.

Kind regards,

Nafisa M. Jadavji, PhD, MSc, BSc

Academic Editor

PLOS One

Journal Requirements:

Reviewers' comments:

Reviewer's Responses to Questions

**Comments to the Author**

Reviewer #1: All comments have been addressed

Reviewer #2: All comments have been addressed

2. Is the manuscript technically sound, and do the data support the conclusions?

Reviewer #1: Yes

Reviewer #2: Yes

3. Has the statistical analysis been performed appropriately and rigorously?

Reviewer #1: Yes

Reviewer #2: Yes

4. Have the authors made all data underlying the findings in their manuscript fully available?

Reviewer #1: Yes

Reviewer #2: Yes

5. Is the manuscript presented in an intelligible fashion and written in standard English?

Reviewer #1: No

Reviewer #2: Yes

Reviewer #1: I have notice some gramatical errors in some part of the manuscript. So, the author better to review and correct it before publication.

Reviewer #2: The revised manuscript has been carefully reviewed. All previously raised comments and requested corrections have been adequately addressed, and no further revisions are required.

.

Reviewer #1: No

Reviewer #2: **Yes:** SALMAN ASHFAQ AHMADSALMAN ASHFAQ AHMADSALMAN ASHFAQ AHMADSALMAN ASHFAQ AHMAD

---

## [Author Response · Author response to Decision Letter 2]

25 Feb 2026

Response to Reviewers

We extend our sincere gratitude to the editors and all reviewers for their valuable comments, which have significantly enhanced the quality of this manuscript. Reviewers' comments are presented in italics, with specific issues numbered. Our responses are indicated in red typeface, and manuscript revisions are highlighted in red.

Comments from Reviewer #1:

1.I have notice some gramatical errors in some part of the manuscript. So, the author better to review and correct it before publication.

Response:Thank you very much for your careful review and for pointing out the grammatical errors in our manuscript. We sincerely appreciate your attention to detail, which has helped us improve the language quality of our work.We have thoroughly reviewed the entire manuscript and corrected all identified grammatical issues. In addition, we engaged a native English speaker with expertise in scientific writing to carefully proofread and polish the language throughout the paper. All changes have been clearly marked in the revised manuscript using the track changes feature to facilitate your re-evaluation.We believe that the revised version now meets the high language standards expected by your journal. We are grateful for your constructive suggestion, which has undoubtedly strengthened the clarity and professionalism of our manuscript.('Revised Manuscript with Track Changes'lines 30-38;68-72;112-115;124-132)

---

## [Editor Report · Decision Letter 2]

22 Mar 2026

Relationship between dietary thiamine, riboflavin, and niacin intake and hypertension subtypes: A cross-sectional study from the 1999-2023

PONE-D-25-55578R2

Dear Dr. Chang,

We’re pleased to inform you that your manuscript has been judged scientifically suitable for publication and will be formally accepted for publication once it meets all outstanding technical requirements.

Kind regards,

Nafisa M. Jadavji, PhD, MSc, BSc

Academic Editor

PLOS One
---

## [Editor Report · Acceptance letter]

PONE-D-25-55578R2

PLOS One

Dear Dr. Chang,

I'm pleased to inform you that your manuscript has been deemed suitable for publication in PLOS One. Congratulations! Your manuscript is now being handed over to our production team.

Kind regards,

on behalf of

Dr. Nafisa M. Jadavji

Academic Editor

PLOS One